# Social and psychological impact of the COVID-19 pandemic on UK medical and nursing students: protocol for a national medical and nursing student survey

George E Richardson [ID],[1] Conor S Gillespie [ID],[2] Orla Mantle,[3] Abigail Clynch,[2] Setthasorn Zhi Yang Ooi [ID],[4] Jay J Park [ID],[5] Emily R Bligh,[6] Shantanu Kundu,[7] Ioannis Georgiou,[8] Soham Bandyopadhyay [ID],[9] Kate E Saunders,[10,11] Neurology and Neurosurgery Interest Group (NANSIG), SPICE-20 Collaborative

GER and CSG contributed equally.

GER and CSG are joint first authors.

For numbered affiliations see end of article.

**Correspondence to**
Dr Conor S Gillespie;
hlcgill2@liv.ac.uk

## ABSTRACT

**Introduction** Healthcare students have played a significant role in the National Health Service during the COVID-19 pandemic. We captured data on the well-being of medical students during the acute phase of the pandemic with the Social and Psychological Impact of COVID-19 on medical students: a national survey Evaluation (SPICE-19) study. We will evaluate changes in mental health and well-being of medical and nursing students 1 year after SPICE-19, in a cross-sectional study, to understand the impact of the pandemic, and inform well-being policies.

**Methods and analysis** This study will be a national, multi-institution, cross-discipline study. An online 53-item survey of demographics, mental health and well-being will be used to record responses. Students studying for a medical or nursing degree at any UK universities will be eligible to participate. The survey will be advertised through the Neurology and Neurosurgery Interest Group national network. Participation is anonymous and voluntary, with relevant mental health resources made available to participants.

**Ethics and dissemination** Ethical approval was granted by the University of Oxford Central University Research Ethics Committee (R75719/RE001) on 21 May 2021. Study findings will be presented at national and international meetings, and submitted for publication in a peer-reviewed journal.

## INTRODUCTION

Medical and nursing students in the UK have been significantly affected by the COVID-19 pandemic. During the first wave, many were active components of the clinical team, either by undertaking extending clinical placements, paid work in the National Health Service (NHS) or through voluntary roles, while others saw placements reduced, suspended or cancelled completely.[1][2] We evaluated medical student well-being and

---

### Strengths and limitations of this study

⇒ This is a multicentre, national questionnaire-based survey of medical and nursing student mental health and well-being in the UK.

⇒ This is the first study to assess mental health and well-being of the student multidisciplinary team during the new normal after the COVID-19 pandemic.

⇒ The study results will provide valuable insight into healthcare workers, which can be used to identify and target further areas of well-being support and inform policy.

⇒ Response bias may overestimate participation of students based on age, sex and medical/nursing school, with those interested in neuroscience and mental health more likely to participate.

---

mental health during the acute phase of the pandemic with the Social and Psychological Impact of COVID-19 on medical students: a national survey Evaluation (SPICE-19) study,[3] a cross-sectional study with a prospective component, which identified several points for service provision improvement based on 2275 responses across 34 medical schools in the UK.[4–6]

It is pertinent to re-evaluate the mental health of medical students, 1 year removed from SPICE-19, to explore how they are navigating their training, education and the impact of adapting to the 'new normal' of medical education provision.[7][8] It is also important to elucidate the effect of COVID-19 on the well-being and mental health of UK nursing students, many of whom were asked to volunteer on the front line in COVID-19 wards,[2] and a large-scale study examining this has not been completed. A single-centre

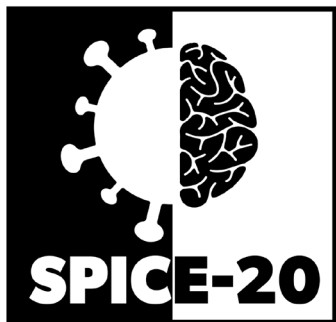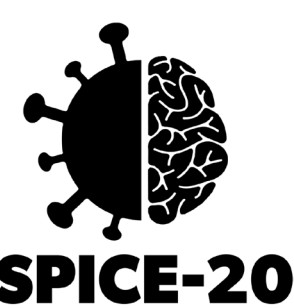

**Figure 1** Social and Psychological Impact of COVID-19 on medical students: a national survey Evaluation (SPICE-20) study logos.

survey in the USA noted that nursing students reported increased levels of anxiety, reduction in concentration and difficulties with academic workload throughout the pandemic,[9] and a large multi-institutional study is required to elucidate this further. This is in accordance with previously published work by the SPICE-19 research team.[10 11]

Medical and nursing schools have each responded differently to the pandemic, with the accessibility of well-being and mental health services variable between universities.[12] A national survey assessing mental health and well-being in medical and nursing students, as well as support systems available for those facing mental health difficulty, is required. Evaluating the mental health and well-being of the student multidisciplinary team is vital for ensuring adequate service provision is provided—it is paramount to safeguard the well-being of the next generation of healthcare professionals.[13 14]

The primary aim of this study is to comprehensively evaluate mental health and well-being of medical and nursing students during the evolving COVID-19 pandemic, 1 year after the initial outbreak. The secondary aims are to elucidate students' perceptions of institutional efforts to provide support and accessibility and appropriateness of such support. The study logos are shown in figure 1.

## METHODS AND ANALYSIS
### Study partners
#### Neurology and Neurosurgery Interest Group
The Neurology and Neurosurgery Interest Group (NANSIG) was formed in 2010. It is a student and junior doctor-led interest group designed to increase participation, diversity and engagement in neurosciences, and has over 1500 members internationally.[15] The organisation has an affiliation with the Society of British Neurological Surgeons (SBNS) and the Association of British Neurologists (ABN) (www.nansig.org). The organisation led the SPICE-19 national cohort study last year, assessing well-being in 2275 medical students in response to the pandemic, and is familiar with survey design and distribution. The collective organisation runs established

national and international events, and has published over 20 peer-reviewed publications.

### Study design
SPICE-20 is a national, multi-institution, cross-sectional study. An online survey will be used to record participant responses. The survey contains 53 items. Section 1 of the survey includes background and demographic information. Section 2 assesses well-being by asking participants to complete the Warwick-Edinburgh Mental Well-Being Scale (WEMWBS); a 14-point validated evaluation of well-being.[16] Section 3 encompasses the Patient Health Questionnaire-9 (PHQ-9) and Generalized Anxiety Disorder-7 assessment (GAD-7). The PHQ-9 is a valid diagnostic and severity measure for depressive disorder,[17] and GAD-7 is a seven-item validated assessment of generalised anxiety.[18 19] Section 4 includes questions about university support offered by each institution, if participants had accessed this support, and any significant changes to teaching encountered during the pandemic. Section 4 contains study-specific unvalidated questions. These questions were defined for the current study to meet the specific secondary aims and were not adapted from the SPICE-19 survey due to a difference in study objectives.

The survey was iteratively defined by the student-led study management team until a consensus was reached. Based on our experience with the SPICE-19 survey, we have reduced the number of questions and limited the survey to validated measures of well-being and mental health. The final 53-item questionnaire is shown in figure 2.

### Survey piloting
The survey was piloted by a group of 15 medical students from the NANSIG core committee. These students were not involved in the design of the study and were consulted in order to provide feedback, improve clarity and ensure objectivity. Data were analysed to identify any points of concern, estimated completion time and difficult survey items/questions. Students were contacted to identify any suggestions for improvement, with none identified. Therefore, no further alterations were made.

### Survey administration
The survey was hosted on the Qualtrics survey platform (Provo, Utah, USA), a General Data Protection Regulation (GDPR)-compliant online survey platform, that facilitates both mobile and desktop devices.

### Study dissemination
To maximise distribution across the UK, a national network of SPICE-20 Collaborative members were recruited, representing all 33 medical schools in the UK, using a purposive sampling method. Most held concurrent committee or regional lead roles in NANSIG at time of recruitment. Each member was asked to acquire the most up-to-date (as of April 2021) available resources, guidance and support policies in place for their current medical school, plus an additional nursing school. A standard advertising

**Section 1 – Background Information**
1. *Are you a medical or nursing student?*
    a. Medical student
        i. Which medical school are you currently enrolled in?
            1. Response from dropdown of medical schools
    b. Nursing Student
        i. Which nursing school are you currently enrolled in?
            1. Response from dropdown of nursing schools
2. How many years (if any) of your degree (excluding intercalation) have you completed so far?
    a. Choose year of study from list
3. Are you currently undertaking an intercalated degree?
    a. Respond yes or no
        i. Have you previously undertaken an intercalated degree?
            1. Respond yes or no
4. How old are you?
    a. Free text box for age
5. What is your gender?
    a. Choose from gender list with free-text box
6. What is your race or ethnicity?
    a. Choose from list of ethnicities with free-text box
7. What is your religion?
    a. Choose from list of religions with free-text box

**Section 2 – Wellbeing and Mental Health**
**WEMWBS**
Participants are shown a series of statements and asked to choose an appropriate response from either: 1) None of the time, 2) Rarely, 3) Some of the time, 4) Often and 5) All of the time
1. "I've been feeling optimistic about the future"
2. "I've been feeling useful"
3. "I've been feeling relaxed"
4. "I've been feeling interested in other people"
5. "I've had energy to spare"
6. "I've been dealing with problems well"
7. "I've been thinking clearly"
8. "I've been feeling good about myself"
9. "I've been feeling close to other people"
10. "I've been feeling confident"
11. "I've been able to make up my own mind about things"
12. "I've been feeling loved"
13. "I've been interested in new things"
14. "I've been cheerful"

**PHQ-9**
Participants are asked, over the last 2 weeks, how often have you been affected by any of the following problems? Responses include: 1) Not at all (0-1 days), 2) Several days (2-6 days), 3) More than half the days (7-11 days), 4) Nearly every day (12-14 days)
1. Little interest or pleasure in doing things?
2. Feeling down, depressed or hopeless?
3. Trouble falling or staying asleep, or sleeping too much?
4. Feeling tired or having little energy?
5. Poor appetite or overeating?
6. Feeling bad about yourself – or that you are a failure or have let yourself or your family down
7. Trouble concentrating on things, such as reading the newspaper or watching television?
8. Moving or speaking so slowly that other people could have noticed. Or the opposite – being so fidgety or restless that you have been moving around a lot more than usual?
9. Thoughts that you would be better off dead or of hurting yourself in some way?

**GAD-7**
Participants are asked, over the last 2 weeks, how often have you been bothered by the following problems? Responses include: 1) Not at all, 2) Several days, 3) More than half the days, 4) Nearly every day
1. Feeling nervous, anxious or on edge
2. Not being able to stop or control worrying
3. Worrying too much about different things
4. Trouble relaxing
5. Being so restless that it is hard to sit still
6. Becoming easily annoyed or irritable
7. Feeling afraid as if something awful might happen

**Other Questions**
1. How would you rate your mood before COVID-19 affected the UK (One year ago)? 0 being the worst mood you can imagine and 100 being the best mood you can imagine
    a. Response via sliding scale from 0 to 100
2. How would you rate your mood now? 0 being the worst mood you can imagine and 100 being the best mood you can imagine
    a. Response via sliding scale from 0 to 100

**Section 3 – Support**
1. Has your university provided any form of mental health and wellbeing support?
    a. Yes
        i. What support has the university provided you with? (please select one or more)
            1. Multiple selection boxes with options and free text box for other
        ii. How confident do you feel accessing the support services available to you?
            1. 5 point Likert scale from not confident at all to very confident
        iii. Have you accessed/used support that your university has provided?
            1. Yes
                a. Has your university support been useful?
                    i. Yes
                        1. In what way?
                            a. Free text response
                    ii. No
                        1. Why not?
                            a. Free text response
            2. No
        iv. Would you like your university to provide you with more wellbeing support?
            1. Yes (please select one or more)
                a. Multiple selection boxes with options and free text box for other
            2. No
                a. Why not?
                    i. Free text response
    b. No

**Section 4 – Changes**
1. Is your teaching different now compared to before the pandemic (do not answer if in first year)?
    a. Yes
        i. Please select one or more of the following ways teaching/placements have changed:
            1. Multiple selection boxes with options and free text box for other
    b. No
        i. Why not?
            1. Free text response
2. Has there been any positive aspects of the changes to teaching?
    a. Yes
        i. Please detail the positive changes to teaching
            1. Free text response
    b. No
3. What changes could be made to teaching in medical school to improve your wellbeing?
    a. Multiple selection boxes with options and free text box for other
4. Have you worked or are you currently working within a healthcare setting during the course of this pandemic?
    a. Yes
        i. Which of the following roles have you taken up?
            1. Multiple selection boxes with options and free text box for other
        ii. Have you been offered wellbeing support in response to the role you have undertaken?
            1. Yes
                a. Who has provided this support? (please select one or more)
                    i. Multiple selection boxes with options and free text box for other
            2. No
    b. No
5. After the pandemic, are you more or less likely to continue in a career in Medicine/Nursing?
    a. 5 point Likert scale from Much less likely to Much more likely
6. Have your future career plans changed since the start of the pandemic?
    a. Yes
        i. If Yes, how come?
            1. Free text response
    b. No

**Figure 2** Final Social and Psychological Impact of COVID-19 on medical students: a national survey Evaluation (SPICE-20) survey design. GAD-7, Generalized Anxiety Disorder-7; PHQ-9, Patient Health Questionnaire-9; WEMWBS, Warwick-Edinburgh Mental Well-Being Scale.

method was developed by the steering committee and based on the successful recruitment strategy employed in the SPICE-19 study, which involved members contacting their respective medical school deans to request distribution of the survey at inception, as well as regular advertising over the 8-week study collection period, from 7 June 2021 to 7 August 2021. The distribution links for the survey include university mailing lists, student society pages and social media platforms. Figure 3 provides an example of the email template used to disseminate the

Dear Students,

We are a group of student researchers that are interested in investigating the current state of medical student mental health and wellbeing, one year on from the onset of the pandemic. We are inviting all UK medical students, from years 1 to 5 (and intercalating) to complete a short 15 minute survey to help us better understand how you are doing. Whilst completing the survey, you will be asked a variety of questions about your mental health and wellbeing, and the support that you have received. Your IP address will not be recorded and responses will be anonymous. There will not be any follow up survey. Useful links to support services have been provided along with the post-survey debrief. Additional information about the risks versus benefits of taking part in the survey can be found by following the study link below. **Please access the survey by the link below:**

https://psychiatryoxford.qualtrics.com/jfe/form/SV_eQmCH0FLewvbcWO

If you have any questions about the study or your involvement please contact a member of the study team at the following email and we will aim to reply within 24hrs:
hlgricha@liverpool.ac.uk

Should you have concerns about your wellbeing do consider contacting you GP or accessing the resources available: <LOCAL SERVICE PROVISION INSERTED HERE>

Many thanks for taking the time to read this,

George Richardson
SPICE-20 Study Team

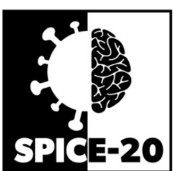 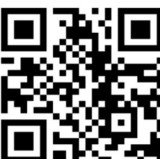 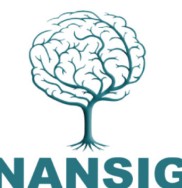

**Figure 3** Standard email template used for all survey disseminations through student email lists, developed by the study team and approved by the research ethics committee. NANSIG, Neurology and Neurosurgery Interest Group; SPICE-20, Social and Psychological Impact of COVID-19 on medical students: a national survey Evaluation.

survey through student mailing lists. Adverts will also be placed on NANSIG social media platforms as well as the monthly NANSIG newsletter.

In the invitation to complete the study, based on their medical/nursing school, participants will also receive well-being, mental health and guidance services specific to their institution, should they need to access them at any point. Participants will also be made aware of our patient and public involvement organisation and their contact details. No incentives for participation will be offered.

### Eligibility and representation

All current students enrolled at UK medical schools recognised by the General Medical Council (GMC) and the Medical Schools Council will be eligible to participate. Additionally, all current home and international students enrolled on an adult, child and mental health nursing degree at a UK university are eligible for inclusion. A list of eligible medical schools and approved programmes by the Nursing and Midwifery Council (NMC) is included in the online supplemental materials.

### Consent and confidentiality

A patient information sheet will be shown on the first page of the study (online supplemental material). This encompasses the rationale, purpose and voluntary nature of participation. Participants must verify that they are over 18 years old and provide informed consent via tick box. It will be emphasised that participation is anonymous, confidential and voluntary, and that participants can withdraw consent at any time.

### Data security

All data will be collected and stored on the secure online server Qualtrics. Survey data will be extracted from the software to a password-protected Microsoft Excel spreadsheet (Microsoft, California, USA) only available to the study team. Data handling and record keeping will adhere to the University of Oxford standards for data security.

### Statistical analysis

Data will be analysed using R V.4.0.3. The overall well-being of medical and nursing students, as assessed using the WEMWBS, and mental health, using the PHQ-9 and GAD-7, will be presented using descriptive statistics. The scores of WEMWBS, PHQ-9 and GAD-7 will be categorised using established cut-off points into dichotomous groups, corresponding to individuals expressing symptoms and those which do not.[20–22] No information that identifies or is specific to institutions will be presented. Appropriate statistical methods will be selected based on the distribution and type of data; it is anticipated data will be non-parametric and will therefore be compared using tests such as $\chi^2$ or Kruskal-Wallis tests. Differences between medical and nursing students will be compared using these statistical tests. To identify factors associated with increased scores on WEMWBS, PHQ-9 and GAD-7, multivariate generalised linear regression will be used.

### Sample size calculation

Approximate estimates of total UK medical and nursing students were calculated from yearly undergraduate intake figures. The total number of UK medical and adult nursing students was estimated to be 84 000. An online calculator was used to determine the sample size needed for the survey. The minimum sample size required from our target population for a margin error of 5% (around a 50% distribution) and a CI level of 95% is 383, assuming a 50% response distribution for each question, the most conservative response distribution for study power.

### Patient and public involvement

As the study population was medical and nursing student well-being and mental health, we worked collaboratively with the Be Free Campaign, a mental health and well-being awareness charity, from study inception. Be Free was involved in the conception, design and development of the study. The charity works with young people and students, emphasising mental health and well-being, to tackle the stigma behind mental health, and promote expression of individual values (UK-registered charity number 1189704). The charity director and clinical study team for the charity comprehensively reviewed the protocol for appropriateness of content, question format,

and ensured the questions asked were sensitive and pertinent to medical and nursing students.

Be Free has extensive knowledge of indicators of mental health, and the questions selected (WEMWBS, PHQ-9 and GAD-7) were done so in concordance with their advice. The number of questions, layout and design were also prepared in accordance with their suggestions. The charity inspired the conception of this study and approved the final survey design. The charity founder and director (SK) is also a member of the steering committee for this study.

## DISCUSSION
### Study rationale

International multi-institutional studies have demonstrated that during the initial COVID-19 outbreak, medical students were less likely to experience deterioration in mental health and well-being when compared with non-medical students.[23 24] This is the first national, multi-institutional study to evaluate the mental health of UK medical and nursing students during the COVID-19 pandemic.[25] It will therefore be possible to elucidate the mental health and well-being of healthcare students across the UK. By comparing medical and nursing schools, it will be possible to evaluate the nature of successful support, as well as students' confidence and ability to access this support. The findings will inform service provision, and highlight areas where improvement is required. The survey will also highlight local and national well-being resources available to students.

### Why assess medical and nursing student well-being and mental health?

Medical and nursing students form the healthcare multidisciplinary team of the future. Mental health and well-being are positively correlated with job satisfaction, retention and performance, and are an essential component of good health.[26] If the NHS is to retain, recruit and develop its workforce of the future, ascertaining their mental health at this point in the COVID-19 pandemic is essential. Young people have been disproportionately affected by the pandemic, and most medical and nursing students fall into this age category. As non-emergency care is delayed and rescheduled, students will have to deal with the long-acting repercussions of the pandemic, including those directly affected by the virus itself, including Long COVID,[27] declining mental health in the population exacerbated by the pandemic,[28] and implications of delayed access to healthcare.[29 30]

Many medical students may not be aware that such support exists, either at their respective institution or at a regional/national level, and we would do these students a disservice by not highlighting existing mental health and well-being services.

### Benefits of a national medical student well-being survey

Medical and nursing schools across the UK adopt heterogenous practices to achieve the outcomes set in the GMC and NMC core competencies and curricula.[31] A survey of both medical and nursing students on a national scale is the best way to evaluate mental health and well-being. Many institutions reacted differently to the pandemic, and a survey is best placed to simultaneously decipher well-being and mental health at each institution, while assessing the impact of the pandemic on a national scale, and highlighting heterogeneity between institutions. Furthermore, there is a need to appreciate that multi-institutional studies of large magnitude are required to answer important clinical questions and elucidate concerns not identified by smaller studies. Multicentre studies are also less susceptible to identity issues because the data are aggregated between multiple medical schools, and can be presented as all centres.

### Limitations

Online surveys have the advantage of ubiquitous presence, and increased accessibility to medical and nursing schools in the UK. However, several limitations can arise from such a study method.

First, those with an existing interest in well-being and mental health may be more likely to engage with and complete the survey. This could lead to selection bias towards those adversely affected by the pandemic. The anticipated magnitude of this effect is unclear. In addition, as the survey is being distributed through NANSIG, there may be overparticipation from students with an interest in neurosciences and their surrounding connections.

In order to ameliorate this as best as possible, we have adopted a varied dissemination approach. This includes having representatives at each UK medical school, survey distribution to medical and nursing school bulletins, and newsletters. We hope this will improve visibility and participation. We decided not to incentivise participants with a potential prize for completing the survey, as we wanted responses to be motivated by a desire to complete the questions in as open a way as possible.

The study is also limited by its cross-sectional design, and will therefore only provide a snapshot of mental health and well-being at the time of survey completion. In addition, during capricious times such as COVID-19, fluctuations in mental health may occur in accordance with policy changes, as well as institutional policies towards placements and assessment, and we will assess date of study completion as a potential confounder as a variable in our analysis.

The self-reporting of symptoms will also be a significant limitation, in addition to the fact that the questions included are screening tools for mental health and well-being, and are not immediately diagnostic. Furthermore, a number of important variables including socioeconomic status, comorbid psychiatric conditions and levels of isolation have not been included within the survey.[32 33] These factors are known to be associated with adverse mental health at baseline in young healthcare students,[34] and finances have been noted specifically to contribute to adverse medical and nursing student well-being.[32] The

reason we omitted these from the survey, but included other variables such as ethnicity and religion, was for two reasons. First, it was felt that, during the conception of the study, including religion and ethnicity as part of the participant demographics section was feasible, and would not affect participants' responses. Second, it was felt that requesting participants to divulge finances and report ongoing social isolation would be too intrusive, and is not a common component of participant demographics for most surveys. Omission of these and other variables may cause confounding of study results. The decision not to include these variables was made to ensure minimal barriers existed to completion of the survey; however, we understand that the results must be interpreted within the context of this limitation.

Finally, there are no embedded security or validation measures to confirm the medical or nursing school of each participant, and therefore we cannot guarantee that all participants were UK medical or nursing students.

### Ethics and dissemination

Ethical approval for this study was granted by the Oxford Research Ethics Committee (R75719/RE001). The findings of the SPICE-20 study as described in this protocol will be presented at scientific conferences, and submitted for publication in a peer-reviewed journal once finalised.

**Author affiliations**
[1]Institute of Translational Medicine, University of Liverpool School of Medicine, Liverpool, UK
[2]Institute of Systems, Molecular and Integrative Biology, University of Liverpool, Liverpool, UK
[3]Faculty of Life Sciences and Medicine, King's College London, London, UK
[4]Cardiff University School of Medicine, Cardiff, UK
[5]The University of Edinburgh Medical School, Edinburgh, UK
[6]School of Medicine, The University of Sheffield, Sheffield, UK
[7]School of Medicine, University of Liverpool, Liverpool, UK
[8]School of Medicine, University of Aberdeen, Aberdeen, UK
[9]Oxford University Medical School, University of Oxford, Oxford, UK
[10]Department of Psychiatry, University of Oxford, Oxford, UK
[11]Oxford Health NHS Foundation Trust, Oxford, UK

**Acknowledgements** The authors would like to thank the SPICE-19 Collaborative, whose work inspired this study, and are grateful to the Be Free Campaign for their assistance with the conception, design and inspiration for components of the study. We also thank NANSIG and members of the SPICE-20 Collaborative for their kind assistance disseminating the survey.

**Collaborators** Neurology and Neurosurgery Interest Group (NANSIG): Conor S Gillespie, Soham Bandyopadhyay, Setthasorn Zhi Yang Ooi, Jay Jaemin Park, Alvaro Yanez Touzet, Emily R Bligh, George E Richardson, Abigail Clynch, Abdullah Egiz, Seong Hoon Lee, Oliver Burton, William Bolton, Bharti Kewlani, Moritz Steinruecke, Avani Shanbhag, Joshua Erhabor, Orla Mantle. SPICE-20 Collaborative: George E Richardson, Conor S Gillespie, Orla Mantle, Abigail Clynch, Anita Golash, Anjali Deepak, Anson Wong, Catincia Ciuculete, Cora Lowe, Dana Hutton, Electra Lerou, Emily Bligh, George Davis, Gideon Adegboyega, Hannah Redpath, Hanya Ghazi, Jacob Porter, Jashan Selvakumar, Jay Park, La-Dantai Henriques, Mc Stephen S Padilla, Mehdi Khan, Mohammed Talha Bashir, Ollie Burton, Prithvi Bahu, Robyn Wilcha, Rohan Gupta, Rosaline de Koning, Ritika Dilip, Setthasorn Zhi Yang Ooi, Shankari Gnanakumar, Shantanu Kundu, Sumaiya Rizaam, Tom Hess, Tomas Ferreira, Ioannis Georgiou, Soham Bandyopadhyay, Professor Kate E Saunders.

**Contributors** GER, CSG and OM were responsible for initial drafting of protocol manuscript, conceptualisation and designing the study. AC, SZYO, JJP, ERB, IG, SB and KES formed the wider study group and reviewed and approved the drafts of the manuscript. KES provided supervision of the project. SK reviewed the manuscript and also provided PPI and approval for the study.

**Funding** CSG is a recipient of a grant from the Wolfson Foundation. GER is the recipient of funding from North West Cancer Research. KES is supported by the NIHR Oxford Health Biomedical Research Centre.

**Disclaimer** The views expressed are those of the author(s) and not necessarily those of the NHS, the NIHR or the Department of Health.

**Competing interests** None declared.

**Patient and public involvement** Patients and/or the public were involved in the design, or conduct, or reporting, or dissemination plans of this research. Refer to the Methods section for further details.

**Patient consent for publication** Not required.

**Provenance and peer review** Not commissioned; externally peer reviewed.

**ORCID iDs**
George E Richardson http://orcid.org/0000-0001-5610-451X
Conor S Gillespie http://orcid.org/0000-0002-9153-3077
Setthasorn Zhi Yang Ooi http://orcid.org/0000-0002-7097-0948
Jay J Park http://orcid.org/0000-0001-8762-6986
Soham Bandyopadhyay http://orcid.org/0000-0001-6553-3842

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
