## [Reviewer comments · BMJ Open]

ARTICLE DETAILS

TITLE (PROVISIONAL)	Social and Psychological Impact of the Covid-19 pandemic on UK medical and nursing students: Protocol for a national medical and nursing student survey
AUTHORS	Richardson, George; Gillespie, Conor; Mantle, Orla; Clynch, Abigail; Ooi, Setthasorn Zhi Yang; Park, Jay; Bligh, Emily R.; Kundu, Shantanu; Georgiou, Ioannis; Bandyopadhyay, Soham; Saunders, Kate E; Interest Group, Neurology and Neurosurgery; SPICE-20, Collaborative

VERSION 1 – REVIEW

REVIEWER	Htay , Mila University of Malaya, Department of Social and Preventive Medicine
REVIEW RETURNED	15-Oct-2021

GENERAL COMMENTS	Abstract 1. The abstract is concise and well-described about the protocol of the study.2. In the abstract, the authors mentioned as “We will evaluate changes in mental health and wellbeing of medical and nursing students one year after the study in order to understand the ongoing impact of the pandemic and inform wellbeing policies.”. The study design is described as a cross-sectional study. This sentence seems that it is a continuation of the SPICE 19, longitudinal design. It would be better if the authors clarify for this point. Introduction 3. The authors well-described the background of the study and objectives. Methods 4. Regards to the sample size calculation, the expected response distribution was assumed as 50%. I am just wondering if the researchers could have used the finding of responses from the SPICE 19 study. May I know if there is any justification for using 50% of the expected frequency in this study?5. Please include the sampling method of this study.6. It would be better if the authors include the planning about the scoring and classification of the study tool (WEMWBS, GAD-7, PHQ-9) in the protocol.7. I would suggest the authors describe more about section 4, whether it is adapted from the previous study, any validation process conducted for this section, etc.
---

REVIEWER	Wathelet, Marielle Centre Hospitalier Regional Universitaire de Lille, Psychiatry
REVIEW RETURNED	17-Dec-2021

GENERAL COMMENTS	Thank you for asking me to review this protocol. This study will monitor the mental health of nursing students and medical students, one year after the start of the Covid-19 pandemic, as recommended. Indeed, the Lancet's COVID-19 Commission Mental Health Task Force recommends monitoring the mental health of populations over the next few years¹. 1 Aknin, Lara B., Jan E. De Neve, Elizabeth W. Dunn, Daisy Fancourt, Elkhonon Goldberg, John Helliwell, Sarah Jones, et al. 2021. "Mental Health During the First Year of the COVID-19 Pandemic: A Review and Recommendations for Moving Forward." PsyArXiv. February 19. doi:10.31234/osf.io/zw93g. The protocol is well written, and the authors present several guarantees of quality and feasibility:  - The SPICE-20 study is a national, multi-institution, cross-sectional study. However, the SPICE-19 study (including 2,275 students) could be used as a reference to describe the evolution of the disorders. - The experience of SPICE-19 demonstrates the feasibility of the project. - The team involves students in the piloting of the study, which ensures that the survey is adapted to the target population. - The tools used are validated scales. - It is an online survey, however, to maximize distribution across the UK, a national network includes members representing all 33 medical schools in the UK. - Resources are offered to students in the event that they present mental health issues I have a few questions:  - I see that there is no financial incentive, which can be a security, but have other means been implemented to ensure that the respondents are indeed medical students or nursing students? Otherwise, a limit could be added. - I do not see, in the questionnaire, questions evaluating the financial situation of the student, his/her psychiatric history nor level of isolation. However, these are factors associated with the presence of disorders. In addition, socio-economic conditions can vary between medical students and nursing students, which may constitute a confounding factor. Could you justify? Although the precaution on this subject varies from one country to another, for what reasons should religion and ethnicity have been prioritized? - Regarding the rationale, it seems important to refer to the results of studies comparing medical or nursing students to other students. For example, two Chinese studies found that medical students were less likely to suffer from distress, severe anxiety, and depression than non-medical students during the initial stage of the 2019 coronavirus disease (Chang et al., 2020¹; Xie et al., 2020²). These results were confirmed by a national multi-institution study, carried out in France, and including nearly 70,000 students (Leroy et al. 2021³). 1 Chang, J., Yuan, Y. and Wang, D. (2020) 'Mental health status and its influencing factors among college students during the epidemic of COVID-19', Nan fang yi ke da xue xue bao = Journal
--

	of Southern Medical University, 40(2), pp. 171–176. doi: 10.12122/j.issn.1673-4254.2020.02.06. 2 Xie, L. et al. (2020) 'The immediate psychological effects of Coronavirus Disease 2019 on medical and non-medical students in China', International Journal of Public Health, 65(8), pp. 1445–1453. doi: 10.1007/s00038-020-01475-3. 3 Leroy A, Wathelet M, Fovet T, Habran E, Granon B, Martignère N, Amad A, Notredame CE, Vaiva G, D'Hondt F. Mental health among medical, healthcare, and other university students during the first COVID-19 lockdown in France. J Affect Disord Rep. 2021 Dec;6:100260. doi: 10.1016/j.jadr.2021.100260. Epub 2021 Oct 31. PMID: 34746911; PMCID: PMC8557945. - The authors describe their calculation of sample size. However, the authors assume a prevalence of 50%. From the results of SPICE-19 or the study by Leroy et al., the authors could define the lowest expected prevalence (and adjust the level of precision accordingly). The calculation of the sample size from this hypothesis seems more appropriate. - There is one element that I did not quite understand. The authors say that for every medical student response, 1.28 nursing student responses would be needed to achieve a 50:50 distribution in response rate. Could the authors clarify, and describe what this 1.28 refers to? - Multivariate analyzes are not described. They could be briefly specified.
--	--

VERSION 1 – AUTHOR RESPONSE

Reviewer 1

“The abstract is concise and well-described about the protocol of the study.”

1. Thank you for this comment.

“In the abstract, the authors mentioned as “We will evaluate changes in mental health and wellbeing of medical and nursing students one year after the study in order to understand the ongoing impact of the pandemic and inform wellbeing policies.”. The study design is described as a cross-sectional study. This sentence seems that it is a continuation of the SPICE 19, longitudinal design. It would be better if the authors clarify for this point.”

2. Thank you for this- we have now changed the abstract, to clearly indicate that this study was a cross sectional design. (line 5 and 6, introduction)

“The authors well-described the background of the study and objectives.”

3. Thank you very much for this comment.

“Regards to the sample size calculation, the expected response distribution was assumed as 50%. I am just wondering if the researchers could have used the finding of responses from the SPICE 19 study. May I know if there is any justification for using 50% of the expected frequency in this study?”

4. This was based on, at the time of study inception, not knowing the full results of the SPICE-19 study. Therefore, we could not assume the response distribution, and selected 50%, which results in

the largest sample size possible, for power calculations. This is the most conservative option, and ensures that any number exceeding the sample size would be more likely to represent appropriate power.

“Please include the sampling method of this study”

5. This has now been included in the methods section (study dissemination, line 3)

“It would be better if the authors include the planning about the scoring and classification of the study tool (WEMWBS, GAD-7, PHQ-9) in the protocol.”

6. Thank you for highlighting this. We have now addressed this point in the statistical analysis section (line 3), complete with relevant references.

“I would suggest the authors describe more about section 4, whether it is adapted from the previous study, any validation process conducted for this section, etc.”

7. Thank you for your comment. We have expanded on the decision for including unvalidated study specific questions within section 4 and detailed this in the study design paragraph (paragraph 1, from line 10).

Reviewer 2:

“This study will monitor the mental health of nursing students and medical students, one year after the start of the Covid-19 pandemic, as recommended. Indeed, the Lancet’s COVID-19 Commission Mental Health Task Force recommends monitoring the mental health of populations over the next few years¹.”

8. Thank you for this comment. We agree with the reviewers, that this is a highly important issue, and should be explored by studies such as the one we have designed.

“The protocol is well written, and the authors present several guarantees of quality and feasibility”

9. Thank you very much for this comment.

“I see that there is no financial incentive, which can be a security, but have other means been implemented to ensure that the respondents are indeed medical students or nursing students? Otherwise, a limit could be added.”

10. Indeed, there was no actual security measures to confirm whether or not respondents were medical students or nursing students. We have added this as a limitation to discussion, page 10, paragraph 5, lines 3-5

“I do not see, in the questionnaire, questions evaluating the financial situation of the student, his/her psychiatric history nor level of isolation. However, these are factors associated with the presence of disorders. In addition, socio-economic conditions can vary between medical students and nursing students, which may constitute a confounding factor. Could you justify? Although the precaution on this subject varies from one country to another, for what reasons should religion and ethnicity have been prioritized?”

11. We would like to thank the reviewer for raising this concern. We recognise that our survey is not all-encompassing and there are important variables which have been omitted, and therefore may introduce confounding of study results. The decision to omit such factors was made on the basis of keeping overall study items to a minimum and ensuring the most convenient to answer (but still relevant) questions were included, especially given the high degree of survey fatigue noted following the Covid-19 outbreak. To address this, we have highlighted this consideration in the limitations section.

“Regarding the rationale, it seems important to refer to the results of studies comparing medical or nursing students to other students. For example, two Chinese studies found that medical students were less likely to suffer from distress, severe anxiety, and depression than non-medical students during the initial stage of the 2019 coronavirus disease (Chang et al., 2020; Xie et al., 2020). These results were confirmed by a national multi-institution study, carried out in France, and including nearly 70,000 students (Leroy et al. 2021).”

12. Thank you for this comment. We agree that it is important to set the research in the proper context. As such, we have included the relevant references to provide context related to mental health and wellbeing of medical vs non-medical students.

“The authors describe their calculation of sample size. However, the authors assume a prevalence of 50%. From the results of SPICE-19 or the study by Leroy et al., the authors could define the lowest expected prevalence (and adjust the level of precision accordingly). The calculation of the sample size from this hypothesis seems more appropriate.”

13. We recognise that this comment requires addressing. A similar comment was raised by the previous reviewer and as such we have addressed this within the manuscript and the previous comment (Reply #4). Many thanks.

“There is one element that I did not quite understand. The authors say that for every medical student response, 1.28 nursing student responses would be needed to achieve a 50:50 distribution in response rate. Could the authors clarify, and describe what this 1.28 refers to?”

14. We apologise for any confusion caused by this statement. The intent behind this point was to illustrate that due to the size difference in potential participants (There are more nursing students in the UK than medical students), to achieve a 50:50 ratio of respondents, more nursing students were needed to reply than medical students. We recognise that this is perhaps a somewhat redundant statement and have therefore removed it from the manuscript.

“Multivariate analyses are not described. They could be briefly specified.”

15. We have expanded on this within the statistical analysis section of the protocol. Thank you for highlighting this.

VERSION 2 – REVIEW

REVIEWER	Wathelet, Marielle Centre Hospitalier Regional Universitaire de Lille, Psychiatry
REVIEW RETURNED	22-Feb-2022
GENERAL COMMENTS	I thank the authors for their corrections.

	1. As requested, the authors have added a limitation relating to the verification of the identity of the participants: “Finally, there are no imbedded security measures to confirm a respondents medical or nursing school, and therefore we cannot guarantee that all participants were UK medical or nursing students.” [typo: “a respondents”] 2. The authors confirm the absence of collection of confounding variables such as socio-economic variables or psychiatric history. This lack of information remains problematic but is now cited as one of the limitations. “Furthermore, a number of important variables including socio-economic status, co-morbid psychiatric conditions, and levels of isolation, have not been included within the survey. Omission of these and other variables may cause confounding of study results. The decision not to include these variables was made to ensure minimal barriers existed to completion of the survey, however we recognise that the results must be interpreted within the context of this limitation”. [typo: “recognize”] 3. Insofar as the authors did not directly answer the questions addressed, I do not have their answers concerning the question on the collection of ethnicity and religion. Their collection does not seem justified enough. 4. As advised, the authors have added the results of studies comparing students according to their major: “International multi-institutional studies have demonstrated that during the initial COVID-19 outbreak, medical students were less likely to experience deterioration in mental health and wellbeing when compared to non-medical students.” 5. The subject calculation needed for prevalence studies is such that effectively, setting 50% is a conservative assumption for a fixed precision, at 5% for example (50% [45%-55%]). However, if the expected prevalence is much lower, the precision may no longer be adequate (5% error around 5% is relatively less “precise” than 5% error around 50%). If the authors confirm their desire to maintain their precision at 5%, the sample size is correct. 6. Analyzes have been briefly described.
--	---

VERSION 2 – AUTHOR RESPONSE

Reviewer: 2

Comments to the Author:

- I thank the authors for their corrections.

Thank you very much for this comment, we are grateful for the comprehensive review of the manuscript.

- As requested, the authors have added a limitation relating to the verification of the identity of the participants: “Finally, there are no imbedded security measures to confirm a respondents medical or nursing school, and therefore we cannot guarantee that all participants were UK medical or nursing students.” [typo: “a respondents”]

Thank you very much for this comment. We have now corrected the “a respondents” typing error (changed to “participants”).

- The authors confirm the absence of collection of confounding variables such as socio-economic variables or psychiatric history. This lack of information remains problematic but is now cited as one of the limitations. “Furthermore, a number of important variables including socio-economic status, co-morbid psychiatric conditions, and levels of isolation, have not been included within the survey. Omission of these and other variables may cause confounding of study results. The decision not to include these variables was made to ensure minimal barriers existed to completion of the survey, however we recognise that the results must be interpreted within the context of this limitation”. [typo: “recognize”]

Thank you for highlighting our corrections, we have also corrected the typing error for “recognise” (changed to “understand”).

- Insofar as the authors did not directly answer the questions addressed, I do not have their answers concerning the question on the collection of ethnicity and religion. Their collection does not seem justified enough.

Our apologies for not addressing this specifically in the first round of revisions. We have directly addressed these concerns, specifically covering the inclusion of ethnicity and religion, in the updated limitations section of the manuscript.

4. As advised, the authors have added the results of studies comparing students according to their major:

“International multi-institutional studies have demonstrated that during the initial COVID-19 outbreak, medical students were less likely to experience deterioration in mental health and wellbeing when compared to non-medical students.”

Thank you for highlighting our corrections.

5. The subject calculation needed for prevalence studies is such that effectively, setting 50% is a conservative assumption for a fixed precision, at 5% for example (50% [45%-55%]). However, if the expected prevalence is much lower, the precision may no longer be adequate (5% error around 5% is relatively less “precise” than 5% error around 50%). If the authors confirm their desire to maintain their precision at 5%, the sample size is correct.

Thank you for this. The authors do wish to confirm a precision rate of 50% with a 5% error.

6. Analyzes have been briefly described.

Thank you for confirming our corrections. We feel these now make the manuscript for publication in BMJ Open.